# OpenReview forum: "[Proposal-ML] DCRL: Dataset-Constrained Reinforcement Learning for Safe In-Distribution Exploration"
_tsinghua.edu.cn/THU/2024/Fall/AML — THU 2024 Fall AML Submission_

### Official Review · ~Shaoting_Zhu1 · 2024-11-06
**Review of submission 29**

**Rating:** 8
**Confidence:** 4

**Review:**

The paper introduces a novel approach in the field of offline reinforcement learning (RL) that addresses the critical issue of out-of-distribution (OOD) actions. The authors propose the Dataset-Constrained Reinforcement Learning (DCRL) framework, which integrates safe RL theory into offline RL by leveraging the Dataset Feasibility Function (DFF). This function assesses the feasibility of state-action pairs within dataset constraints, guiding policy learning to respect dataset boundaries while managing OOD risks. The DCRL framework employs two mechanisms: Dataset Feasibility Guidance (DFG) and Dataset Feasibility Indication (DFI), which serve as a regularization term and an OOD detection tool, respectively. The approach aims to blend safety constraints with regularization and counterfactual reasoning to enhance performance and robustness in offline RL.

**Strength**
1. Innovative Integration of Offline RL and Safe RL Theory: The paper presents a pioneering integration of safe RL principles directly into the offline RL OOD problem, providing a structured approach to mitigate OOD risks and promote safe policy learning.
2. Clear task and metric: At the end of the proposal, the author plans to test the algorithm in four different datasets/agents.
3. Detailed survey and method: The paper has a very detailed method with preliminaries, theorems, and pseudo codes. And the related works and baselines to compare are clearly proposed.

**Weakness**
It's too long for a proposal. The page limit is severely exceeded. This leads to the core method being hard to catch, and there is no figure to illustrate the method. Based on this detailed proposal, I think the author may have already done many works on this project, but the majority of the article is not feasible for proposal.

**Questions**
The proposed method relies highly on datasets. However, in many cases, we have no sufficient dataset. For example, the legged robot Go1, the popular method is to train from scratch, and we will have no data before it can walk. I think it is like the problem "The chicken or the egg first." Also, the performance of the final policy highly depends on the performance of the dataset because the algorithm always finds the nearest action in the dataset. This may be missing some of the original purpose of reinforcement learning, which is to learn by exploring. This pattern can make it harder to develop new strategies such as extreme locomotion (jumping to a high box) of the legged robot.

---

### Official Review · ~Yinuo_Li1 · 2024-11-06
**Clear problem definition and solid research**

**Rating:** 9
**Confidence:** 3

**Review:**

This proposal presents a promising approach to safe offline RL by integrating a dataset feasibility function to manage OOD risks. The methodology is sound, and empirical results are strong. Comprehensive review of related works are given and very detailed method and experiments are showed.

Further discussion on computational complexity and generalization to other domains would be beneficial.

---

### Official Review · ~Zhen_Leng_Thai1 · 2024-11-08
**Detailed and Innovative Proposal with Strong Concepts**

**Rating:** 8
**Confidence:** 3

**Review:**

This paper presents a novel Dataset-Constrained Reinforcement Learning (DCRL) framework, featuring Dataset Feasibility Guidance (DFG) for policy regularization and Dataset Feasibility Indication (DFI) for OOD detection—both valuable contributions to offline reinforcement learning. Additional rationale for selecting legged robots or toy car tracking as testing scenarios would be beneficial. However, the paper exceeds the two-page proposal requirement by an additional seven pages, violating submission rules.

---

### Official Review · ~Yanchen_Wu1 · 2024-11-08
**Great research problem and solid theoretical basis**

**Rating:** 8
**Confidence:** 5

**Review:**

In this paper, the author integrates safe learning theory into offline reinforcement learning to make up for the weakness of offline reinforcement learning. In the appendix, the author also shows solid theoretical preparation and algorithm design. I am very confident that this will be a very good research.

---

### Official Review · ~Xiying_Huang2 · 2024-11-09
**Innovative Dataset-Constrained Approach Enhances Safety in Offline Reinforcement Learning**

**Rating:** 8
**Confidence:** 4

**Review:**

This paper introduces a novel Dataset-Constrained Reinforcement Learning (DCRL) framework aimed at improving safety and effectiveness in offline reinforcement learning by constraining out-of-distribution (OOD) actions. The approach employs Dataset Feasibility Guidance (DFG) and Dataset Feasibility Indication (DFI) mechanisms, which guide policy learning within safe dataset boundaries. Extensive benchmark testing demonstrates that DCRL outperforms traditional methods in both safety and policy efficacy.
Quality:
The paper is technically sound, presenting a well-structured approach to enhancing safety in offline reinforcement learning.

Clarity:
The authors provide clear explanations of the DCRL components and empirical results that validate the framework’s effectiveness.

Originality:
The use of feasibility functions for safe OOD exploration marks a novel contribution to offline RL research.

Significance:
DCRL shows strong potential for applications in safety-critical domains, such as robotics and autonomous systems.

Pros:
	•	Novel approach with a strong safety focus in offline RL.
	•	Comprehensive empirical support showcasing improved performance and safety.
	•	Potential impact on real-world applications in risk-sensitive environments.

Cons:
	•	Feasibility function dependency on pre-existing datasets could limit adaptability.
	•	Further discussion on computational requirements and parameter adjustments would enhance reproducibility.

---

### Official Review · ~Xun_Wang10 · 2024-11-10
**Review for "DCRL: Dataset-Constrained Reinforcement Learning for Safe In-Distribution Exploration"**

**Rating:** 9
**Confidence:** 4

**Review:**

This paper introduces a Dataset-Constrained Reinforcement Learning (DCRL) framework for offline RL, which combines safe RL principles with regularization and OOD detection mechanisms to ensure safe and effective policy learning within dataset constraints.

Strength: The proposal thoroughly explains the problem background, related work, and algorithm principles. The experiments are also carefully designed. Moreover, the idea of applying safe RL theory to the offline RL domain is highly innovative.

Weakness: For a proposal, the length and certain content may be somewhat excessive.

---

### Official Review · ~Gausse_Mael_DONGMO_KENFACK1 · 2024-11-11
**Innovative and Good Research Work**

**Rating:** 8
**Confidence:** 4

**Review:**

This paper introduces Dataset-Constrained Reinforcement Learning (DCRL), a framework that integrates safe RL principles into offline RL. The main innovation is the Dataset Feasibility Function (DFF), which helps guide policy learning within dataset boundaries, addressing out-of-distribution (OOD) risks inherent in offline RL. The DFF is implemented via two complementary strategies: Dataset Feasibility Guidance (DFG), which regularizes policies to stay within dataset constraints, and Dataset Feasibility Indication (DFI), an OOD detection mechanism allowing controlled exploration of feasible but riskier regions.

strengths : The dual mechanism of DFG and DFI is well-formulated, presenting a novel way to manage OOD actions. The approach is validated on multiple benchmarks.

weakness: The proposal is too long and does not really respect the guidelines.

---

### Official Review · ~Zhixuan_Pan1 · 2024-11-12

**Rating:** 9
**Confidence:** 3

**Review:**

This proposal introduces Dataset-Constrained Reinforcement Learning (DCRL) for safe policy learning in offline RL. By implementing a Dataset Feasibility Function (DFF), the approach limits policy exploration within dataset boundaries while managing out-of-distribution (OOD) actions to improve both safety and performance.

Pros:

1.Innovative integration of safe RL into offline RL, enhancing policy robustness within constrained environments.
Dual mechanism (DFG and DFI) effectively balances safe exploration with performance in offline RL tasks.

Cons:

1. This is more like a well-developed project rather than a proposal for new work.

2. The optimization process after introducing constraints seems consistent with the standard Lagrange multiplier method. However, this is neither mentioned in the related work section nor addressed in the theoretical analysis.

---

### Official Review · ~Zihan_Wang7 · 2024-11-12
**Review for "DCRL"**

**Rating:** 9
**Confidence:** 3

**Review:**

**Summary:**
DCRL introduces safe reinforcement learning theory into offline RL, proposing the Dataset Feasibility Function (DFF) to manage OOD risks. Through Dataset Feasibility Guidance and Dataset Feasibility Indication mechanisms, it achieves safe exploration under dataset constraints and demonstrates superior performance on D4RL benchmarks.

**Highlights:**
- Novel integration of safe RL and offline RL theories
- Introduction of DFF to evaluate state-action pair feasibility
- Dual-mechanism design (DFG/DFI) for safe exploration

**Advice:**
Given the comprehensive experimental results already present, suggest focusing on enhancing the experimental section, add ablation studies to analyze individual contributions of DFG and DFI.

---

### Official Review · ~Suraj_Joshi2 · 2024-11-12
**Excellent Proposal**

**Rating:** 10
**Confidence:** 4

**Review:**

Everything perfect, my only concern is that there might some avalanche effect due when one agent makes an error that error would be propagated to other agents and hence enlarged, besides that well articulated....all the best!! Excited to see project in action!!